# Drug trends and harm related to new psychoactive substances (NPS) in Sweden from 2010 to 2016: Experiences from the STRIDA project

**Anders Helander** [1,2]*, **Matilda Bäckberg**[3], **Olof Beck**[1,2,4]

1 Department of Laboratory Medicine, Karolinska Institutet, Stockholm, Sweden, 2 Clinical Pharmacology, Karolinska University Laboratory, Stockholm, Sweden, 3 Swedish Poisons Information Centre, Stockholm, Sweden, 4 Department of Clinical Neurosciences, Karolinska Institutet, Stockholm, Sweden

* anders.helander@ki.se

**Data Availability Statement:** All relevant data are within the manuscript and its Supporting Information files.

## Abstract

### Background

In the past decade, hundreds of new psychoactive substances (NPS) have been introduced as unclassified alternatives to the illicit drugs. The NPS represent a growing health concern by causing adverse effects and deaths but are usually undetectable by conventional drug tests. This report summarizes results and experiences from analytically confirmed drug-related acute intoxications in emergency departments (ED) and intensive care units (ICU) enrolled in the Swedish STRIDA project on NPS in 2010–2016.

### Methods and findings

ED/ICU intoxications suspected to involve NPS were enrolled in the project, after initial contact with the Poisons Information Centre (PIC). Serum/plasma and urine samples, and sometimes drug products, were subjected to a comprehensive toxicological investigation, and the PIC retrieved information on associated clinical symptoms and treatment. Between January 2010-February 2016, 2626 cases were enrolled. The patients were aged 8–71 (mean 27, median 24) years and 74% were men. Most biological samples (81%) tested positive for one, or more (70%), psychoactive drugs, including 159 NPS, other novel or uncommon substances, classical recreational and illicit drugs, and prescription medications. When first detected, most NPS or other novel substances (75%) were not banned in Sweden, but they usually disappeared upon classification, which however often took a year or longer. Some NPS were found to be especially harmful and even fatal.

### Conclusions

The STRIDA project provided a good overview of the current drug situation in Sweden and demonstrated a widespread use and rapid turnover of many different psychoactive substances. The accomplishment of the project can be attributed to several key factors (close collaboration between the PIC and laboratory to identify suspected poisonings, free

**Funding:** This work was supported by the Public Health Agency of Sweden (Folkhälsomyndigheten). The funder had no role in study design, data collection and analysis, decision to publish, or preparation of the manuscript.

**Competing interests:** The authors have declared that no competing interests exist.

analysis, continuous updating of analytical methods, evaluation of adverse effects, and sharing information) that are useful for future activities addressing the NPS problem. The results also illustrated how drug regulations can drive the NPS market.

## Introduction

New psychoactive substances (NPS) that are not covered by current drug legislation (previously commonly called "designer drugs") have emerged since the 1960s. In the mid-2000s, herbal smoking mixtures sold under the brand name Spice and producing cannabis-like effects, although not containing Δ-9-tetrahydrocannabinol (THC), became available. After discussions on Internet drug chat forums in 2006 regarding which herbal ingredients were accountable for the psychoactive effects, it was demonstrated in 2008 that the products had in fact been laced with synthetic cannabinoid receptor agonists (SCRA) originally developed as therapeutic drug candidates and pharmacological tools to probe the endocannabinoid system [1–3]. The SCRA additives were banned in some countries but then soon replaced by other, uncontrolled structural variants [4].

This turned out to be a starting point for the ongoing global NPS era, whereby novel substances often bearing only small structural differences are continuously introduced through open trading on the surface web (hence also called "Internet drugs") to replace those becoming controlled [5]. As a result, there has been a dramatic increase in the supply and availability of drugs intended for recreational use in the past decade [6]. The NPS initially comprised mainly SCRA and stimulants (e.g. cathinones), but later also hallucinogenic, sedative, analgesic and dissociative drugs appeared. The European Monitoring Centre for Drugs and Drug Addiction (EMCDDA) is currently monitoring more than 700 NPS that have been notified since 1997 to the European Union (EU) Early Warning System (EWS) operated by the EMCDDA and Europol [6].

The NPS represent a growing health concern, because there is usually no or only limited information available regarding their pharmacological and toxicological dose-response effects, thereby increasing the risk for adverse events and death. Indeed, emergency departments (ED) and intensive care units (ICU) are experiencing many severe poisonings involving NPS [7–9]. For example, analogues of fentanyl have required immediate medical support but also caused many fatalities due to the potentially life-threatening symptoms of opioid overdose [10–12], but unexpected acute and late toxic effects have also been reported for several other novel substances [13–15]. Furthermore, there are analytical problems to confirm NPS exposure, because the new substances are often undetectable by conventional toxicology tests, or may generate false-positive screening results for classical illicit drugs due to close structural similarities [16–18].

In Sweden, a nationwide project called STRIDA was started in 2010, to monitor the occurrence and health hazards of NPS appearing in the country, through evaluation of analytically confirmed serious adverse events among patients requiring emergency hospital care [7, 8]. The aim of this report was to summarize, evaluate and disseminate the results of the ~6-years STRIDA project on NPS, and consider how the generated experiences could be useful in future activities addressing the NPS problem.

## Materials and methods

### Selection of clinical cases and data acquisition

The STRIDA project (the name is an acronym of the Swedish project name, but the word also means "to fight") was run in 2010–2016 as a collaboration between the Karolinska Institutet, the Karolinska University Laboratory, and the Swedish Poisons Information Centre (PIC) in

Stockholm. The PIC is a nationwide 24/7 telephone consultation service concerning acute intoxications operated by pharmacists and medical doctors and is open for health care professionals and the public. All ED and ICU in the country were informed about the project by letter and whenever the PIC was consulted regarding suspected NPS- or unknown drug-related overdose cases. The ED/ICU also received a project laboratory request form that allowed free-of-charge comprehensive toxicological analysis in included cases.

During the planning of the STRIDA project, the PIC organized an internal NPS working group, as the number of consultations related to NPS or unknown drugs had shown a marked increase since 2007 while remaining constant for the classical illicit drugs [14]. Once a new drug was indicated to the project from PIC telephone inquiries, or notified from other sources of information (the project team had continuous communication and information sharing about the NPS situation with the Public Health Agency of Sweden, the Swedish National Forensic Centre, the National Board of Forensic Medicine, the Customs laboratory, the Medical Products Agency, as well as with the EMCDDA/EU EWS), the PIC documentation and treatment guidelines were updated and brief reports on case progress and recovery generated, which meant providing up-to-date toxicological information on new drugs. The Karolinska University Laboratory, who performed the toxicological analyses, was also informed, to allow for updating of analytical methods.

Patients admitted to ED/ICU after exposure to NPS or related products, or where poisoning by such substances was suspected after clinical investigation, met the inclusion criteria for participation in the STRIDA project. Participation required that the medical staff contacted the PIC to obtain a case code number to ensure patient anonymity and submitted biological samples for drug testing and a related laboratory request form containing clinical and treatment information and any other information relevant to the case. During the PIC consultation, case notes on the age, gender, symptoms, and treatment were recorded, and patient self-reports on the substance and/or branded name used, the dose, time of intake, and route of administration were collected, when available.

Urine and serum/plasma samples should be collected as soon as possible after arrival in the ED/ICU, according to standard routines for drug testing, and forwarded to the Karolinska University Laboratory, Department of Clinical Pharmacology in Huddinge, Stockholm. In the laboratory, an aliquot of the urine sample was immediately taken for analysis, as detailed below. The remaining urine volume and the serum/plasma samples were stored frozen until further used.

The clinical and treatment information from the ED/ICU and in de-identified hospital medical records shared with the PIC were paired with the analytical results through the code number. The severity of intoxication was graded retrospectively using the Poisoning Severity Score (PSS) [19], the level of consciousness according to the Reaction Level Scale (RLS) or Glasgow Coma Scale (GCS) [20], and based on the extent of treatment and length of hospital stay. Information on the time for classification of each substance as a narcotic or harmful substance was also collected.

The STRIDA project was conducted in accordance with the Helsinki Declaration and approved by the regional ethics review board (Regionala etikprövningsnämnden) in Stockholm (Nr. 2013/116–31/2).

## NPS drug materials

Drug materials collected by the ambulance personnel or brought to hospital by the patient or accompanying persons were sometimes forwarded to the laboratory, along with the biological samples. The possibility to send drug materials was indicated on the laboratory request form.

The materials were classified as powder, crystal, tablet, capsule, blotter, herbal material, liquid, or other, and stored frozen until subjected for analysis.

In addition, drug products with a claimed content of novel, uncontrolled NPS were sometimes purchased from online vendors, to be utilized as reference material. Following analytical confirmation of the content (see below), the materials were stored frozen.

## Laboratory analysis of biological samples

The biological samples received in the STRIDA project were subjected to a comprehensive toxicological investigation, involving routine immunochemical assays targeting the classical illicit drugs, and multi-component liquid chromatographic–tandem mass spectrometric (LC–MS/MS) and LC–high-resolution single and tandem MS (LC–HRMS(/MS)) screening and confirmation methods targeting NPS, classical drugs, plant- and mushroom-derived substances, and prescription pharmaceuticals, as detailed elsewhere [8, 21–23]. The number of substances covered increased over time, as the LC–HRMS methods were continuously updated when new drugs appeared, and reference materials became available. To demonstrate the presence of newly appearing substances, retrospective evaluation of LC–HRMS data, or reanalysis of biological samples, was sometimes necessary. The lower quantification limit of the MS methods ranged from < 0.5 ng/mL for fentanyls and upwards.

When the analytical investigation of a biological sample was completed, the test result was reported to the ordering clinic.

## Laboratory analysis of drug materials

The analysis of drug materials forwarded to the laboratory along with the biological samples, and of online purchased NPS products, was carried out using LC–HRMS/MS at the Karolinska University Laboratory, or by LC–quadrupole-time-of-flight–MS/MS and nuclear magnetic resonance spectroscopy at the Swedish Medical Products Agency in Uppsala [24, 25]. The investigation provided information on psychoactive substance identity but not on inactive components (e.g., binders in tablets and diluents or fillers in liquids, powders, and capsules), bacteria, viruses, minerals, or metals. Quantitative analysis was only performed for a few products intended for use as preliminary reference materials, until a certified material was available.

## Results

### The STRIDA cases

The recruitment of acute intoxication cases to the STRIDA project (i.e., demographic and clinical data as well as biological samples were obtained) started in January 2010. After only two cases were received in January–February, the number increased to range 5–12 cases per month resulting in a yearly total of 95. In the following years, the number of cases increased markedly to peak at 756 in 2014 (i.e. > 2 cases/day on average) after which it dropped to 659 in 2015 (Fig 1). In 2016, the STRIDA project had to be put on hold due to lack of funding to cover the cost for free toxicological analysis, but still > 100 cases arrived in January–February (Fig 1). During the ~6-years study period, a total of 2626 acute intoxication cases suspected to involve NPS or unknown psychoactive drugs presenting at ED/ICU were enrolled in the STRIDA project. The cases originated from all over Sweden, roughly in proportion to the regional distribution of inhabitants.

During the study period, the PIC experienced a marked increase in consultations related to suspected intoxications by NPS or unknown drugs. The number increased from a yearly starting level around 500 to peak at 1669 in 2014 (Fig 1). These data reveal that STRIDA samples

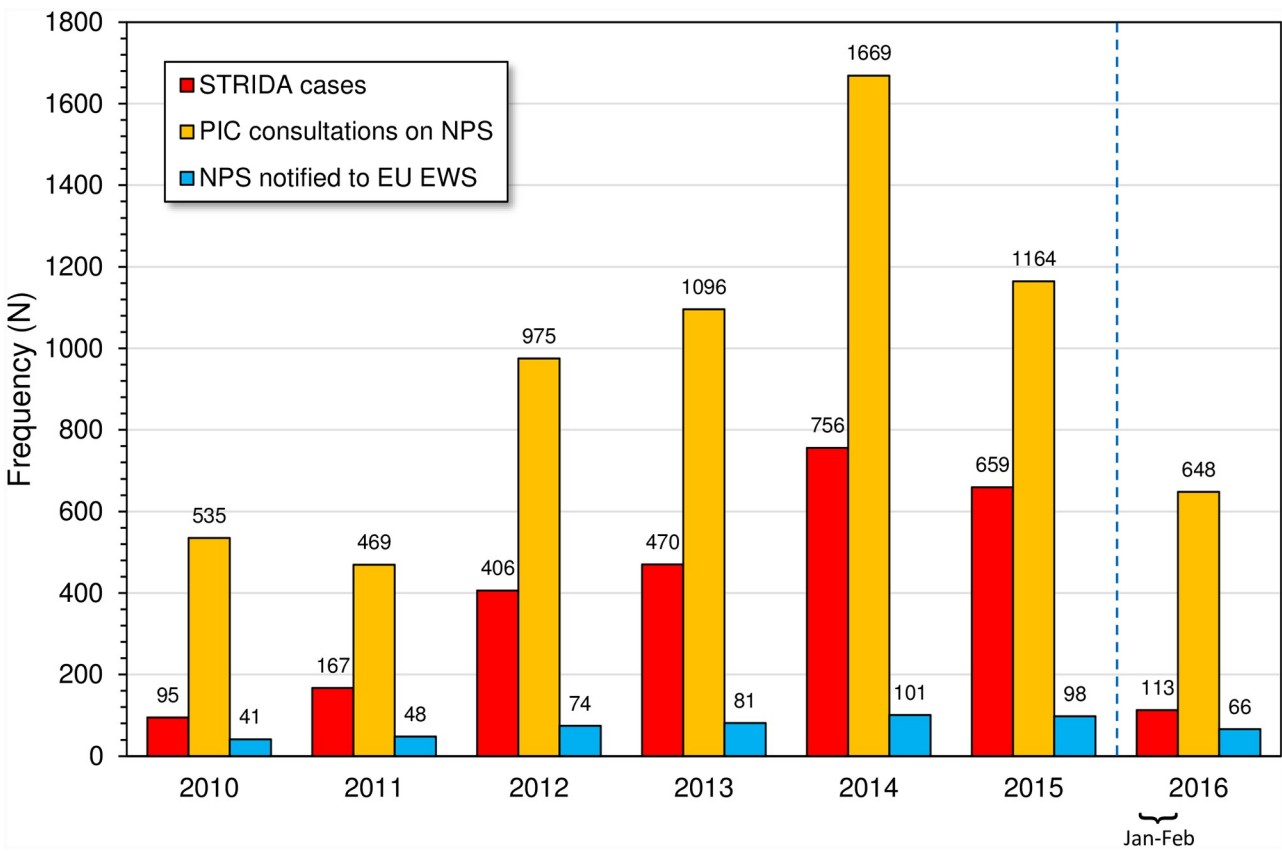

**Fig 1. Statistics on new psychoactive substances (NPS) by year.** Shown are yearly numbers of acute intoxication cases enrolled in the STRIDA project, telephone consultations on NPS with the Swedish Poisons Information Centre (PIC), and NPS reported to the European Union Early Warning System (EU EWS) operated by the European Monitoring Centre for Drugs and Drug Addiction (EMCDDA) and Europol.

were sent to the laboratory in slightly less than half of all PIC consultations suspected to involve NPS (the average for 2010–2015 was 40%). However, the relative proportion showed a steady increase with time and in 2015, the final full year of the project, biological samples were submitted in 57% of the invited cases.

## Demographic data of the STRIDA patients

Overall, 74% of the patients enrolled in the STRIDA project were men but the sex distribution changed slightly over time. The proportion of women was 20% in 2010 but increased to range 24–29% (mean 26.4%) in the following years.

The age range of the STRIDA patients was 8–71 (mean 27.0, median 24) years and 57% were 25 years or younger (Fig 2A). The men were aged 11–72 (mean 27.9, median 25) years which was significantly higher (p < 0.0001; Mann-Whitney test) compared with the 8–60 (mean 24.5, median 21.5) years for women (Fig 2). The overall age of patients increased over time, from a mean (median) age of 22.2 (20) years in the first year to 26.9–28.8 (24–25) years in 2012–2015 (p < 0.0001) (Fig 2B).

Outside the STRIDA project, a newborn girl was tested because her mother was enrolled in the project after taking psychoactive drugs. A urine sample collected on the day after birth tested positive for the cathinone α-PVP (α-pyrrolidinovalerophenone), ritalinic acid (a metabolite of methylphenidate), and ketamine.

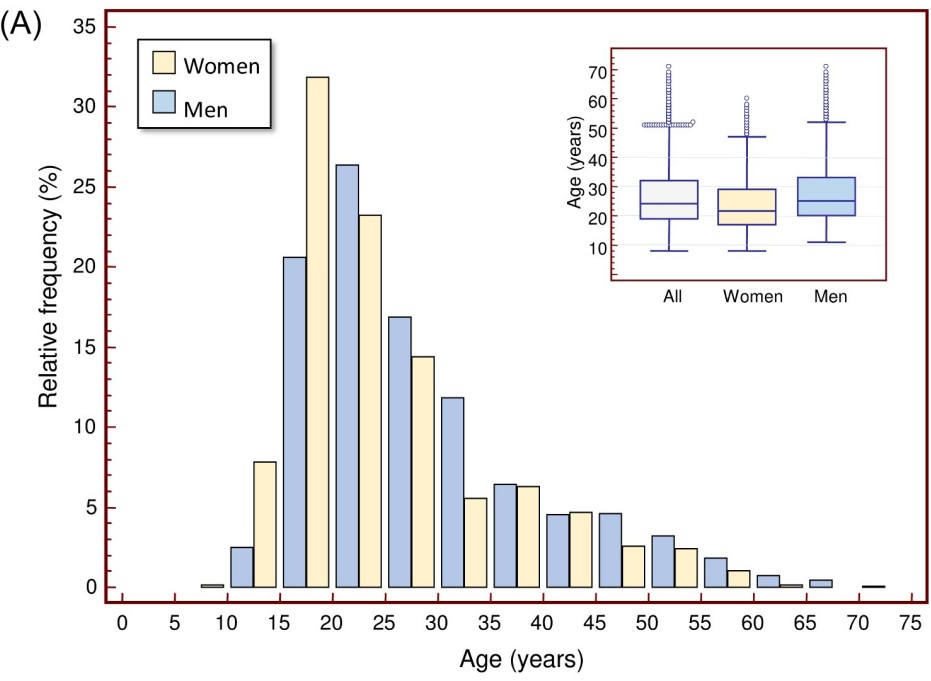

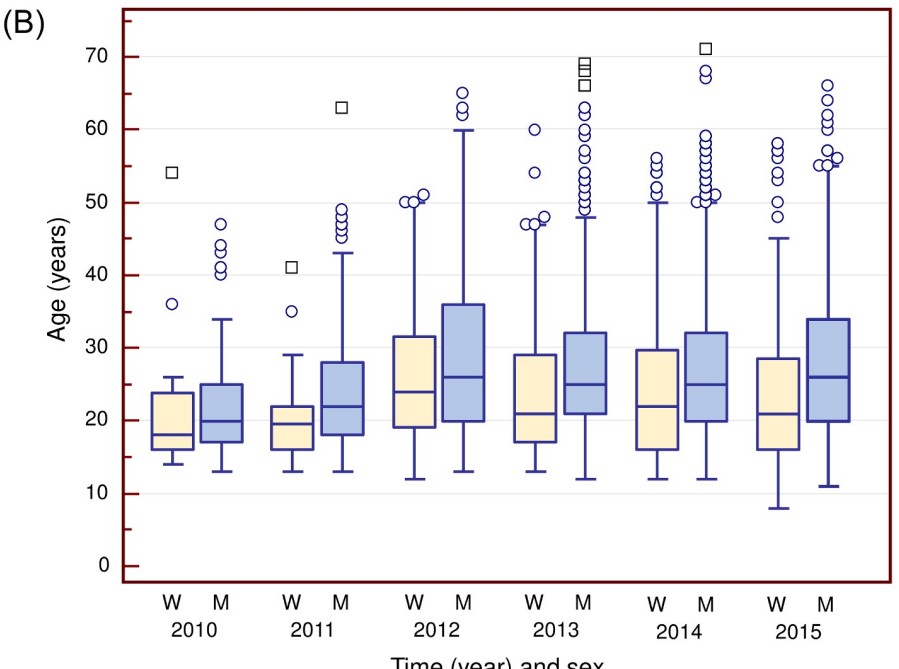

**Fig 2. Age distribution of patients enrolled in the STRIDA project on new psychoactive substances (NPS).** A) Age distribution of male and female STRIDA patients in 2010–2016 (N = 2572). B) Box-and-Whisker plots showing female (W) and male (M) patient ages by year.

## Psychoactive substances detected in the STRIDA cases

The vast majority (81%) of biological samples from ED/ICU submitted for toxicological investigation in the STRIDA project tested positive for psychoactive drugs, and often (70%) for

more than one substance. Detected substances included novel drugs notified by the EU EWS (mainly NPS), classical common (e.g. amphetamine, cannabis and cocaine) and relatively uncommon (e.g. ketamine and psilocybin) illicit drugs in Sweden, approved recreational drugs (ethanol), and prescription medications that are often misused due to their psychoactive effects (e.g. benzodiazepines, buprenorphine and methadone).

From January 2010 until February 2016, a total of 159 novel or less common psychoactive substances were detected in the STRIDA project samples (Fig 3). Most of these (N = 140; 88%) were notified as NPS by the EU EWS in 2004–2016. When first detected, the novel substances were mostly (75%) not yet classified in Sweden (Fig 3).

**New psychoactive substances (NPS).** In 2010, the first year of the STRIDA project, 28 NPS or uncommon drugs were detected in biological samples from the 95 cases (Fig 3), of which 25 substances (89%) were notified for the first time to the EU EWS in 2005–2010 [26]. Stimulants (mainly cathinones) and SCRA (JWH substances) each comprised ~20% of the NPS cases.

In 2011, another 21 novel or uncommon psychoactive substances (Fig 3) were detected in the 167 STRIDA cases (Fig 3), of which 16 (76%) were first notified to the EU EWS in 2005–2012. The NPS continued to be mainly cathinone and amphetamine derivatives.

In 2012, an additional 18 psychoactive substances were detected for the first time in the 406 STRIDA cases (Fig 3), and 14 (78%) of those were notified to the EU EWS in 2004–2015. Besides novel stimulants and SCRA, also designer benzodiazepines (etizolam and metizolam) and hallucinogens (25C- and 25I-NBOMe) started to appear. An especially harmful novel substance was 5-(2-aminopropyl)indole (5-IT) [15], that emerged in Sweden during the first half of 2012 and was involved in at least 15 deaths [27].

In 2013, 21 NPS or uncommon drugs emerged in the 470 cases (Fig 3), of which 19 (90%) were reported to the EU EWS in 2004–2013. Besides additional novel stimulants, cathinones, SCRA, benzodiazepines and hallucinogens, also NPS opioids (AH-7921 and MT-45) and dissociative drugs (3-MeO-PCP) started to appear. MT-45, a former analgesic drug candidate, induced not only typical opioid toxicity but also unexpected severe skin and hearing problems and even blindness requiring surgery [13, 28, 29].

In 2014, 39 novel or uncommon psychoactive substances were detected for the first time in the 756 cases (Fig 3), and 35 (90%) of those were reported to the EU EWS in 2008–2014. The NPS mainly comprised the same set of drug classes as in the previous years, but also butyrfentanyl, a fentanyl analogue without legitimate medical use [30], highly potent SCRA (MDMB-CHMICA) [31], and the phenmetrazine analogue 3-fluorophenmetrazine [32], all being linked to serious toxicity.

In 2015, 27 NPS or uncommon drugs were detected for the first time in the 659 STRIDA cases (Fig 3), and all but one (96%) were reported to the EU EWS in 2006–2016. Additional harmful designer fentanyls (4-fluorobutyrfentanyl, acetylfentanyl, 4-methoxybutyrfentanyl and furanylfentanyl) [30, 33] and benzodiazepines (e.g. clonazolam and nifoxipam) [34] appeared, and also novel hallucinogens (escaline and derivatives).

Finally, during the first months of 2016, after which the STRIDA project was put on hold, a few additional novel drugs appeared that were reported by the EU EWS in 2010–2016 (Fig 3).

In general, each NPS appeared for a relatively short period of time and commonly disappeared upon legal classification (Fig 3). In a retrospective comparison, however, there was a clear time delay between the first observation of a new substance in the STRIDA project to the legal response was implemented (median ~1.0 year, mean 1.6 years; range 2 months to ~8 years). Some novel stimulants and cathinones that appeared over several years were exceptions to this, notably MDPV (methylenedioxypyrovalerone) and α-PVP that became most popular after being classified as narcotics (MDPV was detected in 25% of the STRIDA cases in 2012, and in 16% in 2013; α-PVP was detected in ~10% of the cases in 2012–2015) (Figs 3 and 4A).

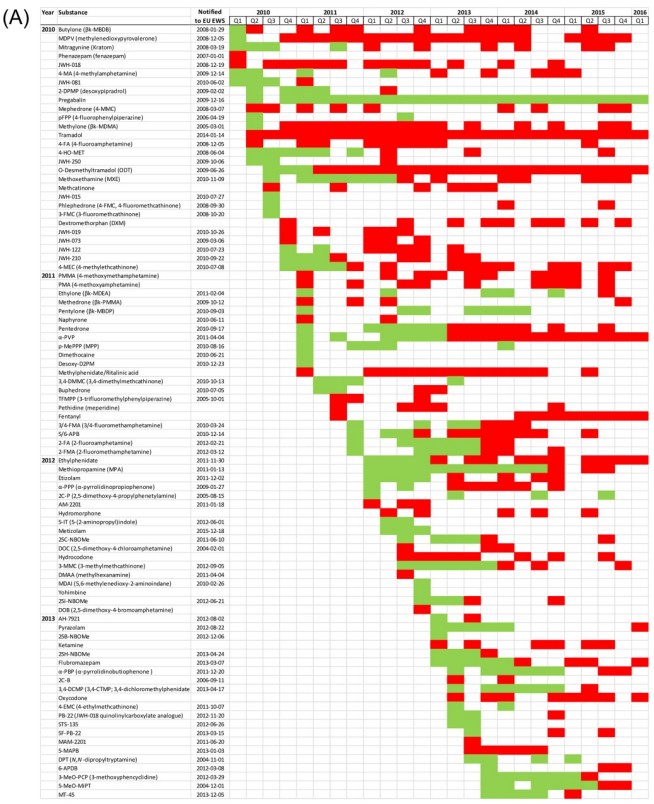

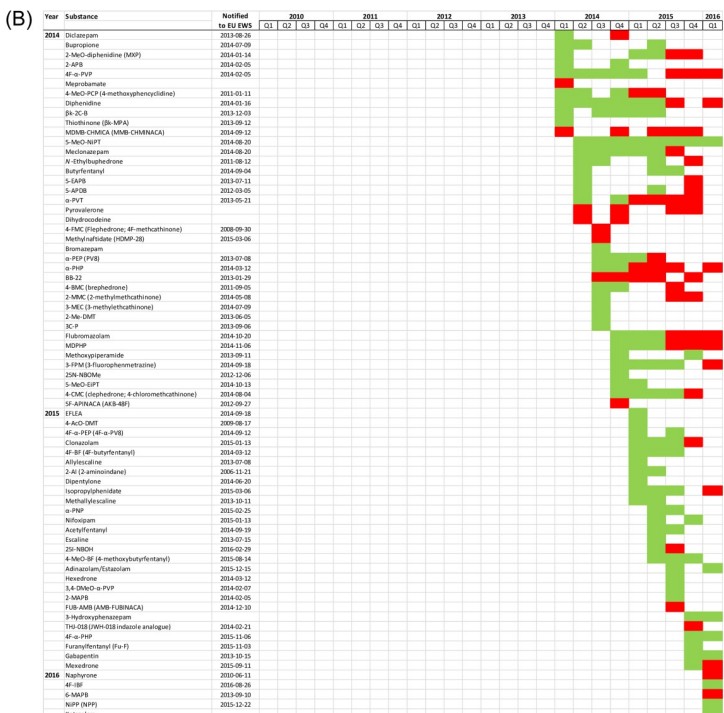

**Fig 3. Novel psychoactive substances detected in the STRIDA project in 2010–2016.** New psychoactive substances (NPS), and other novel or relatively uncommon recreational drugs in Sweden, are listed in order of appearance in the project. The date each substance was notified to the European Union Early Warning System (EU EWS) as an NPS is also given. A green box indicates that when detected the substance was not regulated in Sweden, and a red box that it was classified as either a narcotic or a substance harmful to health.

## (A)                                                      (B)

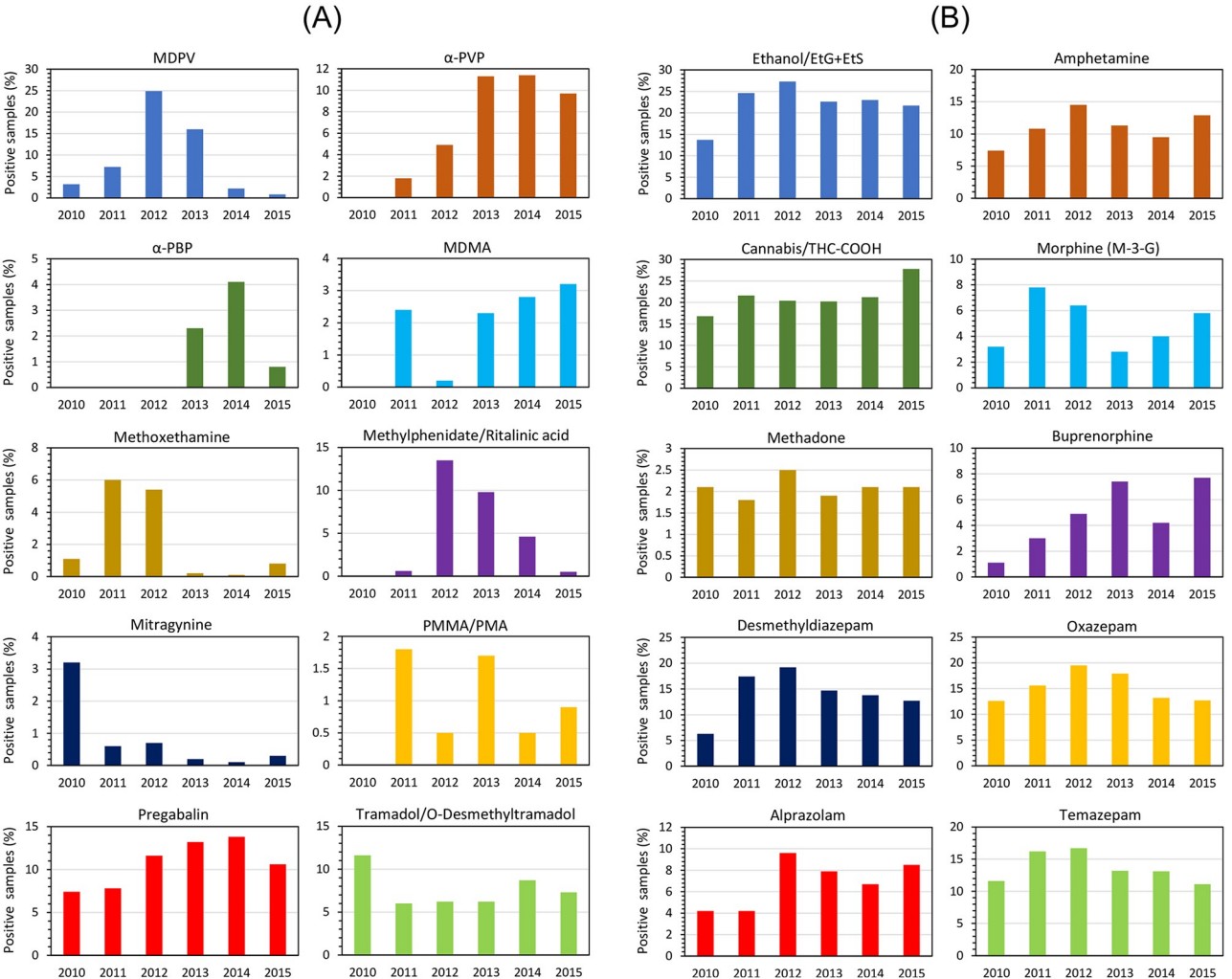

**Fig 4. Yearly frequency of STRIDA project samples testing positive for other common or uncommon psychoactive substances.** The substances included are new psychoactive substances (NPS), other novel or uncommon illicit or unclassified drugs (in Sweden), an approved recreational drug (ethanol), as well as pharmaceuticals with misuse potential.

**Uncommon psychoactive substances.** Psychoactive substances detected in the STRIDA project that have been relatively uncommon as recreational drugs or drugs of abuse in Sweden comprised approved prescription medicines, unapproved medicines, substances controlled internationally under the United Nations Single Convention on Narcotic Drugs of 1961 or the Convention on Psychotropic Substances of 1971, and some other substances that were not banned in Sweden (Fig 4A). Substances of this category that were first detected in 2010 were pregabalin, tramadol and its metabolite O-desmethyltramadol (ODT), and the plant alkaloid mitragynine (kratom). In 2011, also 4-methoxy-amphetamine/-methamphetamine, methyl-phenidate, pethidine (meperidine), and fentanyl appeared. In the following years, additional uncommon drugs detected were 2,5-dimethoxy-4-bromoamphetamine (DOB), hydrocodone, hydromorphone, and the tree alkaloid yohimbine in 2012, ketamine and oxycodone in 2013, and bromazepam, dihydrocodeine, meprobamate and pyrovalerone in 2014 (Fig 3).

Drugs of this category that were demonstrated to be relatively common throughout the years were pregabalin (mean 10.7%, yearly range 7.4–13.8%) and tramadol (mean 7.7%, range 6.0–11.6%) (Fig 4A).

**Classical recreational and illicit drugs.** Besides the many novel and uncommon psychoactive substances detected in the STRIDA cases, also classical recreational drugs (e.g. ethanol) and illicit drugs of abuse (e.g. cannabis and amphetamine) were common analytical findings (Fig 4B). Ethanol and/or its conjugated metabolites ethyl glucuronide (EtG) and ethyl sulfate (EtS) [35, 36], and cannabis (i.e. the metabolite THC-COOH), were found in on average 22.2% (yearly range 13.7–27.3%) and 21.3% (16.8–27.8%), respectively, of all cases (Fig 4B). Ethanol was also the most common single substance intoxication, making up 14% of the positive cases. Medicinal benzodiazepines with a potential of being misused (e.g. diazepam and metabolites) were also common (range for yearly mean 6.9–15.3%) (Fig 4B), but it should be noted that these may sometimes have been administered during the acute handling of patients in the ED/ICU (i.e. before sampling for the STRIDA project). Other common findings were amphetamine (mean 11.1%, yearly range 7.4–14.5%) and buprenorphine (mean 4.7%), whereas cocaine use (i.e. the metabolite benzoylecgonine) was relatively uncommon (mean 1.1%, range 0.3–2.4%) in the study population (Fig 4B).

**Substance use versus age.** When the STRIDA cases were sorted by detected primary substance class (i.e. cannabinoids, ethanol, opioids, stimulants, etc), there was a wide age distribution for patients of all substance categories (S1 Fig). However, patients categorized as cannabinoid or ethanol users had a median age of 20 and 21 years, respectively, which differed significantly ($p < 0.0001$; Mann-Whitney test) from those categorized as mainly opioid, stimulant, dissociative drug, or pregabalin users, who were generally older with median ages of 26–28 years.

## Discussion

The number of acute intoxication cases enrolled in the STRIDA project on the occurrence and health hazards of NPS in Sweden increased markedly with time. Participation in the project was voluntary and submission of biological samples for toxicological investigation largely depended on ED/ICU staff interest and time. A parallel general increased demand for PIC consultations related to NPS, besides for inclusion in the project, indicated a lack of knowledge among hospital care givers about the many novel recreational drugs, which are often named only through combinations of letters and numbers, and how to deal with associated adverse effects. Furthermore, as the test results often were not reported until after discharge of the patients from the ED/ICU, it indicated an interest to contribute to a better awareness and understanding of NPS-related intoxications.

Later during the STRIDA project, the ED/ICU staff routinely contacted the PIC in relevant cases to obtain a project code number and free drug testing, regardless of whether consultation on treatment was needed. When the project could no longer offer free drug testing, due to lack of funding, the number of PIC contacts related to NPS or unknown drugs soon dropped and so did the number of biological samples that were submitted to the laboratory for NPS testing (i.e. routine testing outside the STRIDA project). This indicated that offering free analysis was an important incentive for participation in the project.

Most of the STRIDA patients were 25 years or younger, although the age range was large. The increased age of patients over time may be partly explained by the wider range of substance classes appearing for which the median age of users was higher. In terms of sex distribution, on average 75% were men, but a change was seen over time towards an increased proportion of women.

The analytical results demonstrated a widespread use of many different psychoactive drugs in Sweden in 2010–2016, and polysubstance use was indicated to be common in cases requiring acute hospital care. However, it should be noted that NPS products sometimes contained

two or more psychoactive substances, indicating that all substances detected in the biological samples were not due to intentional intake [24]. In addition to the classical recreational and illicit drugs, with ethanol and cannabis being the most common, 159 novel (mainly NPS) or less commonly used (i.e. in Sweden) psychoactive substances were detected among the 2626 sets of biological samples from acute intoxication cases admitted to ED and/or ICU all over the country. In 2014, when the number of STRIDA cases peaked, it could be estimated that at least 4–5 patients per day on average needed acute hospital care in Sweden due to a suspected NPS-related intoxication, as less than half of all PIC contacts eventually became STRIDA cases. A contributing reason for the peak seen in 2014, when many serious poisonings caused by new potent SCRA appeared [31], may be a major interest in the media for the NPS problem.

For most classical illicit drugs, the relative frequency of positive findings was rather uniform over the years, while there were often highly variable annual figures for most NPS and some of the novel and uncommon substances. The NPS products marketed on Swedish websites and identified in the project also varied over time. Their disappearance often coincided with the date of classification as a narcotic or substance harmful to health which in Sweden is done individually [8]. Once a proposal of classification and the date for implementation was officially notified, the online drug retailers started selling out these substances and replaced them with other unclassified chemical variants [7, 37, 38]. This is evident from the fact that most novel substances (75%) were not classified when first detected in the project samples. Later, however, most of them (92%) became classified as a narcotic or harmful substance.

Exceptions to this were the cathinones MDPV and α-PVP which became most common after being classified [39–41]. However, because MDPV was reported to largely replace amphetamine as the main stimulant drug in some places [42], it is indicated that it was not primarily sold online but through street level dealers. The large number of MDPV and fentanyl analogues introduced over the years also highlights a risk with the individual drug classification system employed in Sweden. Once a new unclassified psychoactive substance has been detected in the country, it must be individually investigated and evaluated for classification as a narcotic or a substance harmful to health, even though it may already be evident from scientific publications to be dangerous [43, 44]. Furthermore, the classification process often takes long time. A retrospective investigation revealed that the median time from first detection of a novel substance in the STRIDA project until classification was one year but sometimes took several years and, meanwhile, these drugs continued to be sold openly. More proactive (generic or analogue) drug classification systems are in place in several other countries [45, 46].

There were sometimes variable annual figures also for the less common psychoactive substances in the STRIDA project. For example, intake of methylphenidate, a classified stimulant medication used to treat attention-deficit/hyperactivity disorder (ADHD), was not found in 2010 and only rarely in 2011, whereas 13.5% of the cases tested positive (i.e. parent drug and/or its metabolite ritalinic acid) in 2012 and close to 10% in 2013. In 2015, however, the frequency dropped to below 1%. Furthermore, in 2010, about 7% of the STRIDA cases tested positive for pregabalin, a medication used to treat epilepsy, neuropathic pain and generalized anxiety disorder, and that figure doubled to 14% in 2014 and remained high in the following years (pregabalin was classified in Sweden in 2018). Methoxetamine, a dissociative hallucinogenic drug, was detected in 5–6% of the cases in 2011–2012 but, after being classified in 2013, in only 1% or less in the other years.

When novel psychoactive substances started to appear, information on their pharmacological and toxicological effects was largely missing, and some later turned out to be more adverse, having very harmful and sometimes fatal effects. Toxicological case series from the STRIDA project highlighting a variety of especially harmful NPS have been published continuously,

covering structural variants of phenethylamine and cathinone stimulants [15, 38–41], halluci-
nogenic and dissociative drugs [37, 47], fentanyl derivatives and other opioids [29, 30, 33, 48,
49], SCRA [31], and designer benzodiazepines [34]. There have also been several bioanalytical
studies related to recommended strategies for drug testing of NPS [16, 17, 23, 50], and results
from the investigation of drug products that were sometimes submitted together with the bio-
logical samples [24].

## Conclusions

The STRIDA project was demonstrated to provide a good overview of the current drug situa-
tion in the country, covering both classical and novel substances. It also served as a warning
system for especially harmful NPS, by sharing collected data on the incidence and distribution,
identification of adverse effects, and treatment of analytically confirmed acute intoxications
presenting in ED and ICU. As such, it represented an important complement to data and sta-
tistics on drug-related deaths. The results of the project also illustrated how drug regulations
can drive the NPS market [51], as the novel substances usually disappeared upon classification
and were replaced by other yet unclassified ones. The STRIDA project supported personnel at
the national PIC, and thereby medical staff all over the country, in the knowledge and critical
care of NPS intoxications, and contributed nationally and internationally through notifications
to the EMCDDA, scientific presentations and publications.

 The accomplishment of the STRIDA project can be attributed to several key factors that
may be considered in future activities addressing the NPS problem:

- A close collaboration between the PIC and the laboratory, to identify suspected cases of NPS
  intoxication from ED and ICU for enrollment in the project, and to follow up on associated
  adverse effects.

- Use and continuous updating of sensitive, preferably HRMS-based, analytical methods to
  follow the changes in drug diversity, as poisonings caused by the latest set of NPS may other-
  wise be hidden.

- Offering free-of-charge toxicological investigation to ensure a high inflow of relevant poi-
  soning cases. Not receiving continued funding to cover the cost for drug testing was the rea-
  son why the project had to be finished.

- Continuous collection of information on the rapid turnover of NPS, through monitoring of
  PIC inquiries, open web sites of drug dealers and drug chat forums, national collaboration
  with e.g. the police and customs, and notifications from the EU EWS.

- Last, but not least, spreading updated information on the ever changing NPS landscape and
  results from the project through lectures and publications.

## Supporting information

**S1 Fig. Box-and-whisker plot showing age distribution of patients in relation to main sub-
stance class.** In 78% of the cases, a classification into either of the following main substance
classes was possible: cannabis or synthetic cannabinoid receptor agonists (Cannabi), ethanol,
hallucinogens (Hallucino), benzodiazepines (Benzodi), dissociative drugs (Dissocia), opioids,
pregabalin (Pregaba), or stimulants (Stimula). N = number of cases; open point = outside
value; filled point = far out value.
(PDF)

## Author Contributions

**Conceptualization:** Anders Helander, Matilda Bäckberg, Olof Beck.

**Data curation:** Anders Helander.

**Formal analysis:** Anders Helander, Matilda Bäckberg, Olof Beck.

**Investigation:** Anders Helander, Matilda Bäckberg, Olof Beck.

**Methodology:** Anders Helander, Matilda Bäckberg, Olof Beck.

**Project administration:** Anders Helander.

**Validation:** Anders Helander.

**Writing – original draft:** Anders Helander.

**Writing – review & editing:** Matilda Bäckberg, Olof Beck.

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
