## [Decision Letter · Decision Letter 0]

12 Feb 2020

PONE-D-19-35565

Drug trends and harm related to new psychoactive substances (NPS) in Sweden from 2010 to 2016: Experiences from the STRIDA project

PLOS ONE

Dear Dr. Helander,

Thank you for submitting your manuscript to PLOS ONE. After careful consideration, we feel that it has merit but does not fully meet PLOS ONE’s publication criteria as it currently stands. Therefore, we invite you to submit a revised version of the manuscript that addresses the points raised during the review process.

Both reviewers found this paper of interest and were consistent in their views on the gaps that need to be addressed in this manuscript. Accordingly, we now invite you to provide point-by-point responses to all comments raised below. 

We would appreciate receiving your revised manuscript by Mar 28 2020 11:59PM. To enhance the reproducibility of your results, we recommend that if applicable you deposit your laboratory protocols in protocols.io, where a protocol can be assigned its own identifier (DOI) such that it can be cited independently in the future. For instructions see: http://journals.plos.org/plosone/s/submission-guidelines#loc-laboratory-protocols

We look forward to receiving your revised manuscript.

Kind regards,

Michelle Tye, Ph.D.

Academic Editor

PLOS ONE

Additional Editor Comments (if provided):

No additional comments.

Journal Requirements:

"The STRIDA project was conducted in accordance with the Helsinki Declaration and approved by the regional ethical review board (Nr. 2013/116–31/2)."

4. Please upload a new copy of Figure 3 A and B as the detail is not clear. Please follow the link for more information: http://blogs.PLOS.org/everyone/2011/05/10/how-to-check-your-manuscript-image-quality-in-editorial-manager/

Reviewers' comments:

Reviewer's Responses to Questions

**Comments to the Author**

1. Is the manuscript technically sound, and do the data support the conclusions?

Reviewer #1: Yes

Reviewer #2: Yes

2. Has the statistical analysis been performed appropriately and rigorously? 

Reviewer #1: N/A

Reviewer #2: N/A

3. Have the authors made all data underlying the findings in their manuscript fully available?

Reviewer #1: Yes

Reviewer #2: Yes

4. Is the manuscript presented in an intelligible fashion and written in standard English?

Reviewer #1: Yes

Reviewer #2: Yes

5. Review Comments to the Author

Reviewer #1: Because of the current NPS epidemic afflicting the US and Europe—with the rapid and steady increase in illicitly-manufactured synthetic drugs, including cathinones, cannabinoids, and synthetic opioids—there is an urgent need for real or near-real time data system that tracks and monitors key indicators associated with drug-involved morbidity and mortality. Such a system would need to incorporate both state and federal government and private institutions so illegal drugs can be track and monitored as they emerge as an illicit product and make their way into illicit supply chains. This manuscript describes a country-wide attempt to do just that: the STRIDA project. Accordingly, the important part of this study, the main and most notable finding, and which is buried as the lead, is the strategy for incorporating government and non-governmental institutions into a communication web that allows for the identification of NPS in a time frame that is conducive for public health response and available as information to people working with drug users and their communities, such as ED docs or medical doctors in the ICU. Yet, as it currently stands, this descriptive information is relatively buried and hidden in the text and the findings isolated by the authors are associated with the identification of drugs using EMODC and PIC to make sense of the toxicological information provided by EDs and ICUs.

All that said, I have numerous comments and suggestions for making the manuscript more centered around STRIDA as a surveillance system that collects information on drugs and associated harms and then takes that information as actionable intelligence that can then be distributed through a centralized repository based on algorithms. I am also supportive of publishing this manuscript without stressing what I think is needed as described above. It’s just that doing so would not contribute all that much to the existing literature on NPS. The U.S. is currently trying to figure out how to get drug-specific information in (near)real time and working to speed up the time frame from when a new drug is discovered at an overdose scene, or seized during an arrest, or collected from people who use drugs by needle exchange. Reconfiguring the paper by stressing the ins and outs of the system (and the findings on drug types and demographics, of course) would make it a manuscript of great interest to public health systems in countries struggling with increasing supply and consumption of NPS, many of which are newly manufactured and making their way into country’s illicit supply chains.

1. The actionable surveillance ‘system’ that I am pointing to could be better described so readers from other countries could understand how STRIDA employed specific tactics for resource sharing and communication between participating institutions of medicine (medical staff), toxicology (Karolinsa/PIC) hospital systems (ED/ICU), poison centers (PIC), and cooperation of international and national organizations (EU EWS/EMD federal drug monitoring programs (STRIDA).

2. The surveillance system described therein is not collecting information and distributing it in what is referred to as real time. I believe the authors report that the timeline for the totality of the process—from the time a drug is collected to when it is identified via toxicology, its pharmacology described, and this information reported back to the parties who initially offered the sample—can take upwards of 8-10 months. So yes, in the context of the existing surveillance systems we have and use in the U.S., and that exist in Europe, may be considered closer to real time since most of the existing systems (NFLIS, NNDSS, SUDORS, TEDS, etc.) can take up to 2 years and sometimes longer to get data. Thus, I have a problem with the authors claiming that STRIDA represents an early warning system that can acquire, analyze, and disseminate information in real-time to be inaccurate. This is a large concern of mine since we’re in need of actionable intelligence related to drug information and associated health effects that can be generated and distributed timelier than 8-10 months. The Conclusion on page 3 describes the system this way and needs to be changed, or at the very least, STRIDA needs to be described in the context of other surveillance systems currently in use to demonstrate that 6-8 months is considered real time that can lead to an early warning.

3. The main research question or questions are unknown. There is no concrete reason for why the study was done. This ambiguity likely contributes to my concerns in #2 because there is nothing to compare the STRIDA system to. It is because most NPS are undetectable on traditional drug screening methods, or is it because of that, of course, but more important is the quality of intelligence gathering on NPS is relatively slow and clunky and the information loops that allow the medical system to acquire information on drugs that may be to blame for medical issues are non-existent or take too long to serve as actionable information. In other words, what is the capacity (sensitivity and specificity) of Sweden’s current drug surveillance systems, how long does it take to identify NPS and associated sequalae, how long does it take for the medical system to get this information, and what mechanisms is information distributed and to whom? These are some of the background information that would make for a productive introduction/lit review, since it would situate what is to demonstrate what could be by highlighting actually existing capacity and the limitation associated with these current systems. The last bullet point is the lead. The study, in my opinion, would be better suited as a description about the process and information algorithms that link the ED/ICU to the PIC and other toxicological generating information so that EDs and ICUs can submit samples with and chief complaints or acquire information on the latest NPS they are seeing in their particular jurisdiction/hospital.

4. We need a system that can do this in a month if we are planning to act on the information. As this study demonstrates, NPS making their way into Sweden’s illicit supply chains occur rapidly and their effects compound even quicker, so it is important that we do not trick ourselves into thinking that a system that takes more than 6 months to generate information useful to EDs and ICUs is a real-time, early warning system. Even SUDORS, CDC’s surveillance system takes several months for the process to complete. They call their timeframe NEAR-REAL TIME, which is perhaps a way to describe STRIDA.

5. How do laboratories fit into the system? On page 8, the authors tell us that the lab was also informed, but they do not say what for.

6. What is the eligible criteria for suspecting NPS are involved in a specific health outcome? The authors tell us on p.8 that much is based on self-report? Was there a form or a universal/standardized way for collecting this information?

7. Why was quantitative analysis only performed on a few product as reference materials (p.9)? Was there no concern for percentages of drugs in samples nor any interest in potency of the product found at a scene, for example? Since labs are used, it would seem useful to have quant findings for a variety of study needs for answering questions related to the correlation between drug and health outcome, much of which is contingent on potency and the percentage of other substance in the sample.

8. Polysubstance use is identified as occurring by urine (and blood?) toxicological analysis but urine only tells you what the decedent or patient ingested but not intent. Put differently, if sellers put myriad psychoactive drugs in a product being sold, could it be that people may be ingesting drugs that they are not taken intentionally? This was the type of situation in the US where fentanyl was being inserted into heroin but unbeknownst to consumers, many of whom did not even want it in their heroin. This is why the testing of drugs found at the scene of an overdose is critical for understanding intent of drugs ingested. Would the authors recommend a different strategy given the limitations associated with urine tox analysis?

Reviewer #2: This paper provide 6 years worth of data about the use of novel psychoactive substances in Sweden, collected through the STRIDA project. The information is based on patients who attend emergency departments and hospitals, where biological samples are analysed for the presences of psychoactive drugs. In some cases drug materials themselves are provided for chemical analysis, too. These data are used to show trends in NPS use over time in relation to when individual NPS were scheduled in Sweden.

Comment on criteria for PLOS ONE:

The study presents primary scientific research which has not been reported elsewhere (although the group have published many papers on individual drugs that have emerged through the project - this paper is the first 2010-2016 omnibus paper from what I can tell). Statistical analyses are descriptive (no inferential statistics are utilised). The data are presented as summary figures and tables and these accord with the claims made in the text and discussion. While described in sufficient detail, the figures need to be in higher resolution to fully assess. The article is intelligible albeit there are some issues with expression (see below). The research has ethics approvals which are satisfactory. Data availability statement is satisfactory.

Comments on the manuscript:

1. The figures are grainy - both in pdf form and in the original form (unless I'm having technical issues). So I found them difficult to properly assess. I recommend they be provided in higher resolution.

2. Demographic data of the STRIDA patients: The last paragraph in this section describes a newborn patient who tested positive, but this case was not included in the range of ages reported in the paragraph before, indicating that the age range was incomplete. The paper should be revised to ensure consistency between this information - either that case is ruled out, or included, and if included, the age range should include this case.

3. In the results: "In the following years, additional less common drugs detected were 2,5-dimethoxy-4-bromoamphetamine (DOB), hydrocodone, hydromorphone, and the tree alkaloid yohimbine in 2012, ketamine and oxycodone in 2013, and bromazepam, dihydrocodeine, meprobamate and pyrovalerone in 2014 (Fig. 3)." Ketamine is not considered a novel substance in Australia, UK, etc. although I am aware that ketamine is still on NPS lists in other places. Is it actually an uncommon or less common substance in Sweden? Perhaps it is worth some explanation for those reading this paper from parts of the world where ketamine is a common recreational substance, in the discussion?

4. In the discussion "More proactive (generic) drug classification systems are in place in several other countries." Please include some citations to articles describing these generic drug classification systems, e.g. van Amsterdam, J., Nutt, D., & van den Brink, W. (2013). Generic legislation of new psychoactive drugs. Journal of Psychopharmacology, 27, 317-324. , and/or Barratt, M. J., Seear, K., & Lancaster, K. (2017). A critical examination of the definition of ‘psychoactive effect’ in Australian drug legislation. International Journal of Drug Policy, 40, 16-25.

5. In the discussion "The median time from first detection of a novel drug in the STRIDA project to classification was ~1.0 year (mean 1.6 years; range 2 months to ~8 years)" - I can't locate this data report in the results section. It would be clearer to have a subsection in the results where the relationship between first detection and classification is described. Related to this, it may be useful for the reader to see more comprehensive aims stated immediately prior to the materials/methods section. This paper not only summarises the results but also charts the emergence of various NPS alongside local legislative actions. Readers interested in drug policy could be alerted to this content earlier, e.g. in the aims.

6. The authors note that this paper was not funded. But it is clear that the STRIDA project itself was awarded funding. Should the source of this funding not be mentioned, as without it, the manuscript would not be possible?

7. There are examples of expression in the manuscript that could be refined. E.g in the abstract 'hundreds of chemically designed new psychoactive substances'. Designed by whom? I'm not entirely sure what 'chemically designed' means. If it means that these NPS are all specifically designed as 'unclassified alternatives to illicit drugs', that may be true for some, but not all - some of these are failed medicines, for example. A further example in the introduction is "Designer drugs aimed at circumventing current drug legislation have occurred since the 1960s". I'm not sure designed drugs have 'occurred' since the 1960s - perhaps they have emerged? Or been designed? A further example: "novel recreational drugs, which are often named through anonymous combinations of letters and numbers". I don't think the authors mean 'anonymous' here?

8. Overall, the abstract is well written and well evidenced. It would improve the abstract if the final sentence 'The accomplishment of the STRIDA project can be attributed to several key factors that can serve as a model for future studies' also listed the 'key factors' other studies could learn from (hopefully just an additional few words).

9. There are some typographical errors in the manuscript, including but not limited to 'trough' in abstract should be 'through'. Please check the manuscript for spelling and grammar errors.

10. In the summary of value of the paper, the authors write "The project served as an effective Early Warning System for harmful NPS, by collecting data on the incidence and distribution, identification of adverse effects, and treatment of analytically confirmed acute intoxications." For the STRIDA project to serve as an effective EWS, it needs to not only collect this information, but also distribute it quickly enough to be useful for a network of stakeholders. Can this sentence be edited to reflect that STRIDA not only collected the information but also distributed it and thereby contributed to an EWS? (note, also applies to the same sentence in the conclusion)

6. PLOS authors have the option to publish the peer review history of their article (what does this mean?). If published, this will include your full peer review and any attached files.

Reviewer #1: Yes: Jon E. Zibbell

Reviewer #2: Yes: Monica Barratt

---

## [Author Response · Author response to Decision Letter 0]

26 Feb 2020

REBUTTAL LETTER – RESPONSE TO REVIEWERS

PONE-D-19-35565

Drug trends and harm related to new psychoactive substances (NPS) in Sweden from 2010 to 2016: Experiences from the STRIDA project

PLOS ONE

• A rebuttal letter that responds to each point raised by the academic editor and reviewer(s). This letter should be uploaded as separate file and labeled 'Response to Reviewers'.

• A marked-up copy of your manuscript that highlights changes made to the original version. This file should be uploaded as separate file and labeled 'Revised Manuscript with Track Changes'.

• An unmarked version of your revised paper without tracked changes. This file should be uploaded as separate file and labeled 'Manuscript'.

Our response: Done.

Additional Editor Comments (if provided):

No additional comments.

Journal Requirements:

Our response: The revised manuscript has been formatted following the style requirements of PLOS ONE. 

"The STRIDA project was conducted in accordance with the Helsinki Declaration and approved by the regional ethical review board (Nr. 2013/116–31/2)."

Our response: The full name of the regional ethics committee (in Swedish) is now given. 

Our response: We have deleted this phrase and instead present the data in a supporting information file (cited as S1 Fig). 

4. Please upload a new copy of Figure 3 A and B as the detail is not clear. Please follow the link for more information: http://blogs.PLOS.org/everyone/2011/05/10/how-to-check-your-manuscript-image-quality-in-editorial-manager/

Our response: New copies of all figures have been generated (and approved) using the PACE system.

Reviewers' comments:

Reviewer's Responses to Questions

Comments to the Author

1. Is the manuscript technically sound, and do the data support the conclusions?

Reviewer #1: Yes

Reviewer #2: Yes

2. Has the statistical analysis been performed appropriately and rigorously?

Reviewer #1: N/A

Reviewer #2: N/A

3. Have the authors made all data underlying the findings in their manuscript fully available?

Reviewer #1: Yes

Reviewer #2: Yes

4. Is the manuscript presented in an intelligible fashion and written in standard English?

Reviewer #1: Yes

Reviewer #2: Yes

5. Review Comments to the Author

Reviewer #1: 

Because of the current NPS epidemic afflicting the US and Europe—with the rapid and steady increase in illicitly-manufactured synthetic drugs, including cathinones, cannabinoids, and synthetic opioids—there is an urgent need for real or near-real time data system that tracks and monitors key indicators associated with drug-involved morbidity and mortality. Such a system would need to incorporate both state and federal government and private institutions so illegal drugs can be track and monitored as they emerge as an illicit product and make their way into illicit supply chains. This manuscript describes a country-wide attempt to do just that: the STRIDA project. Accordingly, the important part of this study, the main and most notable finding, and which is buried as the lead, is the strategy for incorporating government and non-governmental institutions into a communication web that allows for the identification of NPS in a time frame that is conducive for public health response and available as information to people working with drug users and their communities, such as ED docs or medical doctors in the ICU. Yet, as it currently stands, this descriptive information is relatively buried and hidden in the text and the findings isolated by the authors are associated with the identification of drugs using EMODC and PIC to make sense of the toxicological information provided by EDs and ICUs.

All that said, I have numerous comments and suggestions for making the manuscript more centered around STRIDA as a surveillance system that collects information on drugs and associated harms and then takes that information as actionable intelligence that can then be distributed through a centralized repository based on algorithms. I am also supportive of publishing this manuscript without stressing what I think is needed as described above. It’s just that doing so would not contribute all that much to the existing literature on NPS. The U.S. is currently trying to figure out how to get drug-specific information in (near)real time and working to speed up the time frame from when a new drug is discovered at an overdose scene, or seized during an arrest, or collected from people who use drugs by needle exchange. Reconfiguring the paper by stressing the ins and outs of the system (and the findings on drug types and demographics, of course) would make it a manuscript of great interest to public health systems in countries struggling with increasing supply and consumption of NPS, many of which are newly manufactured and making their way into country’s illicit supply chains.

Our response: Thank you for the constructive comments on our manuscript. In the revised version, we have better described the purpose of this report, as well as how different types of information about NPS were collected and the results of the STRIDA project shared. We hope that this is now clearer. 

1. The actionable surveillance ‘system’ that I am pointing to could be better described so readers from other countries could understand how STRIDA employed specific tactics for resource sharing and communication between participating institutions of medicine (medical staff), toxicology (Karolinsa/PIC) hospital systems (ED/ICU), poison centers (PIC), and cooperation of international and national organizations (EU EWS/EMD federal drug monitoring programs (STRIDA).

Our response: To be included in the STRIDA project, ED/ICU clinics treating suspected NPS poisoning cases first needed to contact the Swedish Poisons Information Centre (PIC; the Swedish PIC covers the entire country). When information on a new drug was indicated to the project through PIC telephone consultation or other sources of information (the project had continuous communication and information sharing about NPS with the Public Health Agency of Sweden, the Swedish National Forensic Centre, National Board of Forensic Medicine, Customs laboratory, Medical Products Agency, and the EU EWS), the PIC documentation and treatment guidelines were updated and brief reports on case progress and recovery generated. Accordingly, the PIC could provide up-to-date toxicological information to medical staff on the new drugs appearing in the country. This information is now given in the revised manuscript in the extended first section of the Materials and methods. 

2. The surveillance system described therein is not collecting information and distributing it in what is referred to as real time. I believe the authors report that the timeline for the totality of the process—from the time a drug is collected to when it is identified via toxicology, its pharmacology described, and this information reported back to the parties who initially offered the sample—can take upwards of 8-10 months. So yes, in the context of the existing surveillance systems we have and use in the U.S., and that exist in Europe, may be considered closer to real time since most of the existing systems (NFLIS, NNDSS, SUDORS, TEDS, etc.) can take up to 2 years and sometimes longer to get data. Thus, I have a problem with the authors claiming that STRIDA represents an early warning system that can acquire, analyze, and disseminate information in real-time to be inaccurate. This is a large concern of mine since we’re in need of actionable intelligence related to drug information and associated health effects that can be generated and distributed timelier than 8-10 months. The Conclusion on page 3 describes the system this way and needs to be changed, or at the very least, STRIDA needs to be described in the context of other surveillance systems currently in use to demonstrate that 6-8 months is considered real time that can lead to an early warning.

Our response: As detailed above (#1) and better explained in the revised manuscript, the PIC updated their toxicological consultation documentation on new drugs, when new information was available. As for the laboratory drug test results, these were reported to the ordering clinic as soon as the analysis was completed (also this information is now given), which could take from one or a few days up to several weeks depending on the substance(s) involved. Because this may not be considered as “real-time” and “early warning”, we have excluded these expressions in the revised manuscript. The typically long time from the first appearance of a new drug in the project to its classification as a narcotic or harmful substance (median about 1 year) was, however, never due to a long analysis time but to the legal process.

3. The main research question or questions are unknown. There is no concrete reason for why the study was done. This ambiguity likely contributes to my concerns in #2 because there is nothing to compare the STRIDA system to. It is because most NPS are undetectable on traditional drug screening methods, or is it because of that, of course, but more important is the quality of intelligence gathering on NPS is relatively slow and clunky and the information loops that allow the medical system to acquire information on drugs that may be to blame for medical issues are non-existent or take too long to serve as actionable information. In other words, what is the capacity (sensitivity and specificity) of Sweden’s current drug surveillance systems, how long does it take to identify NPS and associated sequalae, how long does it take for the medical system to get this information, and what mechanisms is information distributed and to whom? These are some of the background information that would make for a productive introduction/lit review, since it would situate what is to demonstrate what could be by highlighting actually existing capacity and the limitation associated with these current systems. The last bullet point is the lead. The study, in my opinion, would be better suited as a description about the process and information algorithms that link the ED/ICU to the PIC and other toxicological generating information so that EDs and ICUs can submit samples with and chief complaints or acquire information on the latest NPS they are seeing in their particular jurisdiction/hospital.

Our response: In the revised manuscript, we have better clarified the purposes of this report at the end of the Introduction. Main aims were to summarize, evaluate and share the results of the �6-year STRIDA project on NPS, and conisder how the experiences generated could be useful for improvement of future activities addressing the NPS problem. Please also see our responses above to issues #1 and #2. 

4. We need a system that can do this in a month if we are planning to act on the information. As this study demonstrates, NPS making their way into Sweden’s illicit supply chains occur rapidly and their effects compound even quicker, so it is important that we do not trick ourselves into thinking that a system that takes more than 6 months to generate information useful to EDs and ICUs is a real-time, early warning system. Even SUDORS, CDC’s surveillance system takes several months for the process to complete. They call their timeframe NEAR-REAL TIME, which is perhaps a way to describe STRIDA.

Our response: As also stated above, we have decided not to use “real-time and early warning” in relation to the STRIDA project. However, the speed at which a new drug can be analytically confirmed, and the analytical and toxicological information shared with the medical and legal community, depends largely on the resources put into the activity, and “near real-time” reporting of analytical results is indeed possible. For example, in a case where nine young men ended up unconscious in the ED/ICU, we were able to identify the involved substance and return the test results to the hospital (and the police) within a few hours after the samples arrived in the laboratory. It turned out the young men had mistaken acrylfentanyl for amphetamine! Acrylfentanyl would not have been detected in any routine hospital laboratory in the country.

5. How do laboratories fit into the system? On page 8, the authors tell us that the lab was also informed, but they do not say what for.

Our response: It is now clarified that all analyses of biological samples in the STRIDA project were performed at the Karolinska University Laboratory. Once the appearance of a new drug was reported or indicated to the project (see also #1), the laboratory started to update the analytical methods and look for reference materials. 

6. What is the eligible criteria for suspecting NPS are involved in a specific health outcome? The authors tell us on p.8 that much is based on self-report? Was there a form or a universal/standardized way for collecting this information?

Our response: An important indication that NPS may be involved, in addition to self-reporting of patients or accompanying persons, was whether patients admitted to the ED/ICU exhibited unknown or unusual symptoms or side effects. Sometimes, NPS drug products were found or were forwarded to the ambulance or medical staff. Furthermore, as already explained in the text (Materials and methods), clinical and other details of the case was given on the project laboratory report form that was attached in each case, in addition to that forwarded during consultation with the PIC, and the possibility to also send drug materials was indicated. 

7. Why was quantitative analysis only performed on a few product as reference materials (p.9)? Was there no concern for percentages of drugs in samples nor any interest in potency of the product found at a scene, for example? Since labs are used, it would seem useful to have quant findings for a variety of study needs for answering questions related to the correlation between drug and health outcome, much of which is contingent on potency and the percentage of other substance in the sample.

Our response: This option was only used in a few cases when certified reference material was not available for purchase or to obtain from other sources (e.g. the National Forensic Centre). This is now better explained in the Materials and methods section under NPS drug materials.

8. Polysubstance use is identified as occurring by urine (and blood?) toxicological analysis but urine only tells you what the decedent or patient ingested but not intent. Put differently, if sellers put myriad psychoactive drugs in a product being sold, could it be that people may be ingesting drugs that they are not taken intentionally? This was the type of situation in the US where fentanyl was being inserted into heroin but unbeknownst to consumers, many of whom did not even want it in their heroin. This is why the testing of drugs found at the scene of an overdose is critical for understanding intent of drugs ingested. Would the authors recommend a different strategy given the limitations associated with urine tox analysis?

Our response: We agree, and this was also an observation from the cited publication on NPS drug materials in the STRIDA project (Bäckberg M, et al. Investigation of drug products received for analysis in the Swedish STRIDA project on new psychoactive substances. Drug Test Anal. 2018;10(2):340-9.). The text has been revised accordingly. The results of a drug test cannot decide the origin of the substance

Reviewer #2: 

This paper provide 6 years worth of data about the use of novel psychoactive substances in Sweden, collected through the STRIDA project. The information is based on patients who attend emergency departments and hospitals, where biological samples are analysed for the presences of psychoactive drugs. In some cases drug materials themselves are provided for chemical analysis, too. These data are used to show trends in NPS use over time in relation to when individual NPS were scheduled in Sweden.

Comment on criteria for PLOS ONE:

The study presents primary scientific research which has not been reported elsewhere (although the group have published many papers on individual drugs that have emerged through the project - this paper is the first 2010-2016 omnibus paper from what I can tell). Statistical analyses are descriptive (no inferential statistics are utilised). The data are presented as summary figures and tables and these accord with the claims made in the text and discussion. While described in sufficient detail, the figures need to be in higher resolution to fully assess. The article is intelligible albeit there are some issues with expression (see below). The research has ethics approvals which are satisfactory. Data availability statement is satisfactory.

Comments on the manuscript:

1. The figures are grainy - both in pdf form and in the original form (unless I'm having technical issues). So I found them difficult to properly assess. I recommend they be provided in higher resolution.

Our response: New copies of all figures have been generated (and approved) using the PACE system. 

2. Demographic data of the STRIDA patients: The last paragraph in this section describes a newborn patient who tested positive, but this case was not included in the range of ages reported in the paragraph before, indicating that the age range was incomplete. The paper should be revised to ensure consistency between this information - either that case is ruled out, or included, and if included, the age range should include this case.

Our response: Although this case was not originally enrolled in the STRIDA project, but only indirectly via her intoxicated mother, we consider the information to be of general interest and suggest maintaining it in the text (side information) but not include it in the age range of STRIDA patients. 

3. In the results: "In the following years, additional less common drugs detected were 2,5-dimethoxy-4-bromoamphetamine (DOB), hydrocodone, hydromorphone, and the tree alkaloid yohimbine in 2012, ketamine and oxycodone in 2013, and bromazepam, dihydrocodeine, meprobamate and pyrovalerone in 2014 (Fig. 3)." Ketamine is not considered a novel substance in Australia, UK, etc. although I am aware that ketamine is still on NPS lists in other places. Is it actually an uncommon or less common substance in Sweden? Perhaps it is worth some explanation for those reading this paper from parts of the world where ketamine is a common recreational substance, in the discussion?

Our response: Ketamine is a relatively uncommon psychoactive substance in Sweden. We have revised the text so that this becomes more obvious to the readers. 

4. In the discussion "More proactive (generic) drug classification systems are in place in several other countries." Please include some citations to articles describing these generic drug classification systems, e.g. van Amsterdam, J., Nutt, D., & van den Brink, W. (2013). Generic legislation of new psychoactive drugs. Journal of Psychopharmacology, 27, 317-324. , and/or Barratt, M. J., Seear, K., & Lancaster, K. (2017). A critical examination of the definition of ‘psychoactive effect’ in Australian drug legislation. International Journal of Drug Policy, 40, 16-25.

Our response: These references were added in the revised manuscript. 

5. In the discussion "The median time from first detection of a novel drug in the STRIDA project to classification was ~1.0 year (mean 1.6 years; range 2 months to ~8 years)" - I can't locate this data report in the results section. It would be clearer to have a subsection in the results where the relationship between first detection and classification is described. Related to this, it may be useful for the reader to see more comprehensive aims stated immediately prior to the materials/methods section. This paper not only summarises the results but also charts the emergence of various NPS alongside local legislative actions. Readers interested in drug policy could be alerted to this content earlier, e.g. in the aims.

Our response: The STRIDA project focused mainly on analytical and acute toxicological issues. After authorities are informed about the presence of a new substance in Sweden, the classification process is handled by the Public Health Agency or, sometimes, the Medical Products Agency. Because the STRIDA project had no impact on this part of the process, we do not consider it to be results of the project and therefore presented these data in the Discussion. 

6. The authors note that this paper was not funded. But it is clear that the STRIDA project itself was awarded funding. Should the source of this funding not be mentioned, as without it, the manuscript would not be possible?

Our response: We agree and will mention the financial sources to the STRIDA project (external: the Public Health Agency of Sweden; internal: Karolinska University Laboratory) when uploading the revised manuscript. 

7. There are examples of expression in the manuscript that could be refined. E.g in the abstract 'hundreds of chemically designed new psychoactive substances'. Designed by whom? I'm not entirely sure what 'chemically designed' means. If it means that these NPS are all specifically designed as 'unclassified alternatives to illicit drugs', that may be true for some, but not all - some of these are failed medicines, for example. A further example in the introduction is "Designer drugs aimed at circumventing current drug legislation have occurred since the 1960s". I'm not sure designed drugs have 'occurred' since the 1960s - perhaps they have emerged? Or been designed? A further example: "novel recreational drugs, which are often named through anonymous combinations of letters and numbers". I don't think the authors mean 'anonymous' here?

Our response: We have checked and revised these expressions and tried to improve the language also in other places. It should be noted that the term “designer drugs” was often used previously for the NPS phenomenon.

8. Overall, the abstract is well written and well evidenced. It would improve the abstract if the final sentence 'The accomplishment of the STRIDA project can be attributed to several key factors that can serve as a model for future studies' also listed the 'key factors' other studies could learn from (hopefully just an additional few words).

Our response: The abstract needed to be shortened considerably to comply with the maximum 300-word limit. Nevertheless, as suggested, we were able to give the key factors in the revised version. 

9. There are some typographical errors in the manuscript, including but not limited to 'trough' in abstract should be 'through'. Please check the manuscript for spelling and grammar errors.

Our response: The revised manuscript has been checked for typos, spelling and grammar. 

10. In the summary of value of the paper, the authors write "The project served as an effective Early Warning System for harmful NPS, by collecting data on the incidence and distribution, identification of adverse effects, and treatment of analytically confirmed acute intoxications." For the STRIDA project to serve as an effective EWS, it needs to not only collect this information, but also distribute it quickly enough to be useful for a network of stakeholders. Can this sentence be edited to reflect that STRIDA not only collected the information but also distributed it and thereby contributed to an EWS? (note, also applies to the same sentence in the conclusion)

Our response: Please see our responses to similar comments given by the first reviewer (issues #2, #4). We do not use the term “early warning” for the STRIDA project in the revised manuscript. 

---

## [Decision Letter · Decision Letter 1]

31 Mar 2020

PONE-D-19-35565R1

Drug trends and harm related to new psychoactive substances (NPS) in Sweden from 2010 to 2016: Experiences from the STRIDA project

PLOS ONE

Dear Dr. Helander,

Thank you for submitting your manuscript to PLOS ONE. After careful consideration, we feel that it has merit but does not fully meet PLOS ONE’s publication criteria as it currently stands. Therefore, we invite you to submit a revised version of the manuscript that addresses the points raised during the review process.

The reviewers both agree that the authors have addressed almost all their comments satisfactorily, there is a minor outstanding comment from Reviewer 2 that remains to be addressed before we can proceed.

We would appreciate receiving your revised manuscript by May 15 2020 11:59PM. To enhance the reproducibility of your results, we recommend that if applicable you deposit your laboratory protocols in protocols.io, where a protocol can be assigned its own identifier (DOI) such that it can be cited independently in the future. For instructions see: http://journals.plos.org/plosone/s/submission-guidelines#loc-laboratory-protocols

We look forward to receiving your revised manuscript.

Kind regards,

Michelle Tye, Ph.D.

Academic Editor

PLOS ONE

Reviewers' comments:

Reviewer's Responses to Questions

**Comments to the Author**

1. If the authors have adequately addressed your comments raised in a previous round of review and you feel that this manuscript is now acceptable for publication, you may indicate that here to bypass the “Comments to the Author” section, enter your conflict of interest statement in the “Confidential to Editor” section, and submit your "Accept" recommendation.

Reviewer #1: All comments have been addressed

Reviewer #2: (No Response)

2. Is the manuscript technically sound, and do the data support the conclusions?

Reviewer #1: Yes

Reviewer #2: Yes

3. Has the statistical analysis been performed appropriately and rigorously? 

Reviewer #1: N/A

Reviewer #2: Yes

4. Have the authors made all data underlying the findings in their manuscript fully available?

Reviewer #1: Yes

Reviewer #2: Yes

5. Is the manuscript presented in an intelligible fashion and written in standard English?

Reviewer #1: Yes

Reviewer #2: Yes

6. Review Comments to the Author

Reviewer #1: (No Response)

Reviewer #2: The authors have addressed my comments to my satisfaction in all but one point, which I comment on below:

Regarding my point 5, "5. In the discussion "The median time from first detection of a novel drug in the STRIDA project to classification was ~1.0 year (mean 1.6 years; range 2 months to ~8 years)" - I can't locate this data report in the results section. It would be clearer to have a subsection in the results where the relationship between first detection and classification is described. Related to this, it may be useful for the reader to see more comprehensive aims stated immediately prior to the materials/methods section. This paper not only summarises the results but also charts the emergence of various NPS alongside local legislative actions. Readers interested in drug policy could be alerted to this content earlier, e.g. in the aims."

Author response: The STRIDA project focused mainly on analytical and acute toxicological issues. After authorities are informed about the presence of a new substance in Sweden, the classification process is handled by the Public Health Agency or, sometimes, the Medical Products Agency. Because the STRIDA project had no impact on this part of the process, we do not consider it to be results of the project and therefore presented these data in the Discussion.

My response: If this is not considered core findings of the study, then the authors should also amend the abstract, because they include the statement in the methods and findings section regarding classification taking a year or longer, and that drugs typically disappeared upon classification. So it is inconsistent to say it is not part of the results of the study, while listing it in the findings section of the abstract. Indeed I think it is one of the most interesting findings of the paper - and therefore it would work best to include the time until classification as a variable in the methods, noting from what data the measure was derived. Then list it as a finding and discuss it as a result and in the discussion. The paper makes a big deal out of it through Figures 3a and 3b, through the colour change green to red, so I think this information is indeed part of the results/findings of the study, even if STRIDA was responsible for the decision to classify. Indeed the final sentence of the abstract "results also illustrated how drug regulations can drive the NPS market" relies in the classification time as part of the results of the paper. So I don't think it makes a lot of sense to relegate this aspect of the paper to a dot point in the discussion.

7. PLOS authors have the option to publish the peer review history of their article (what does this mean?). If published, this will include your full peer review and any attached files.

Reviewer #1: Yes: Jon E. Zibbell

Reviewer #2: Yes: Monica J. Barratt

---

## [Author Response · Author response to Decision Letter 1]

2 Apr 2020

Reviewer #2: The authors have addressed my comments to my satisfaction in all but one point, which I comment on below:

Regarding my point 5, "5. In the discussion "The median time from first detection of a novel drug in the STRIDA project to classification was ~1.0 year (mean 1.6 years; range 2 months to ~8 years)" - I can't locate this data report in the results section. It would be clearer to have a subsection in the results where the relationship between first detection and classification is described. Related to this, it may be useful for the reader to see more comprehensive aims stated immediately prior to the materials/methods section. This paper not only summarises the results but also charts the emergence of various NPS alongside local legislative actions. Readers interested in drug policy could be alerted to this content earlier, e.g. in the aims."

Author response: The STRIDA project focused mainly on analytical and acute toxicological issues. After authorities are informed about the presence of a new substance in Sweden, the classification process is handled by the Public Health Agency or, sometimes, the Medical Products Agency. Because the STRIDA project had no impact on this part of the process, we do not consider it to be results of the project and therefore presented these data in the Discussion.

¨

My response: If this is not considered core findings of the study, then the authors should also amend the abstract, because they include the statement in the methods and findings section regarding classification taking a year or longer, and that drugs typically disappeared upon classification. So it is inconsistent to say it is not part of the results of the study, while listing it in the findings section of the abstract. Indeed I think it is one of the most interesting findings of the paper - and therefore it would work best to include the time until classification as a variable in the methods, noting from what data the measure was derived. Then list it as a finding and discuss it as a result and in the discussion. The paper makes a big deal out of it through Figures 3a and 3b, through the colour change green to red, so I think this information is indeed part of the results/findings of the study, even if STRIDA was responsible for the decision to classify. Indeed the final sentence of the abstract "results also illustrated how drug regulations can drive the NPS market" relies in the classification time as part of the results of the paper. So I don't think it makes a lot of sense to relegate this aspect of the paper to a dot point in the discussion.

Our response: As recommended by the reviewer, we now state in the Methods section that the classification time for novel substances was collected, we have moved the data on median/mean/range time between the first analytical finding in the STRIDA project until the classification became effective from the Discussion to the Results section, and further cover this issue in the Discussion section.

---

## [Editor Report · Decision Letter 2]

7 Apr 2020

Drug trends and harm related to new psychoactive substances (NPS) in Sweden from 2010 to 2016: Experiences from the STRIDA project

PONE-D-19-35565R2

Dear Dr. Helander,

We are pleased to inform you that your manuscript has been judged scientifically suitable for publication and will be formally accepted for publication once it complies with all outstanding technical requirements.

With kind regards,

Michelle Tye, Ph.D.

Academic Editor

PLOS ONE
---

## [Editor Report · Acceptance letter]

10 Apr 2020

PONE-D-19-35565R2 

Drug trends and harm related to new psychoactive substances (NPS) in Sweden from 2010 to 2016: Experiences from the STRIDA project 

Dear Dr. Helander:

I am pleased to inform you that your manuscript has been deemed suitable for publication in PLOS ONE. Congratulations! Your manuscript is now with our production department. 

With kind regards,

on behalf of

Dr. Michelle Tye 

Academic Editor

PLOS ONE